# Diabetes Mellitus: An Independent Risk Factor of In-Hospital Mortality in Patients with Infective Endocarditis in a New Era of Clinical Practice

**DOI:** 10.3390/ijerph16122248

**Published:** 2019-06-25

**Authors:** Cheng-Jei Lin, Sarah Chua, Sheng-Ying Chung, Chi-Ling Hang, Tzu-Hsien Tsai

**Affiliations:** 1Division of Cardiology, Department of Internal Medicine, Kaohsiung Chang Gung Memorial Hospital, No.123, DAPI Rd. Niaosong District, Kaohsiung City 83301, Taiwan; cjlin4@gmail.com (C.-J.L.); chuasr409@hotmail.com (S.C.); miosheny@gmail.com (S.-Y.C.); samuelhang@hotmail.com (C.-L.H.); 2College of Medicine, Chang Gung University, Kaohsiung, Taiwan, No.123, DAPI Rd. Niaosong District, Kaohsiung City 83301, Taiwan

**Keywords:** diabetes mellitus, infective endocarditis, mortality

## Abstract

Infective endocarditis (IE) is a severe disease with a hospital mortality rate of 17–25%. Early identification of IE patients with high risk of mortality may improve their clinical outcomes. Patients with diabetes mellitus (DM) who develop infective diseases are associated with worse outcomes. This study aimed to define the impact of DM on long-term mortality in IE patients. A total of 412 patients with definite IE from February 1999 to June 2012 were enrolled in this observational study and divided into 2 groups: group 1, patients with DM (*n* = 72) and group 2, patients without DM (*n* = 340). The overall in-hospital mortality rate for both groups combined was 20.2% and was higher in group 1 than in group 2 (41.7% vs. 16.5%, *p* < 0.01). Compared to patients without DM, patients with DM were older and associated with higher incidence of chronic diseases, less drug abuse, higher creatinine levels, and increased risk of *Staphylococcus aureus* infection (all *p* < 0.05). Moreover, they were more likely to have atypical clinical presentation and were associated with longer IE diagnosis time (all *p* < 0.05). In multivariable analysis, DM is an independent and significant predictor of mortality. The prognosis of IE patients with DM is still poor. Early identification and more aggressive treatment may be considered in IE patients with DM.

## 1. Introduction

Although infective endocarditis (IE) is a relatively rare cardiovascular disease, it remains a severe disease with a hospital mortality rate of 17% to 25% [1,2,3].

Despite the availability of advanced diagnostic tools and new medications, the mortality rate of IE remains unchanged [1,2,3,4,5]. Therefore, early identification of IE patients with high risk of in-hospital mortality and management of their individual characteristics may improve their clinical outcomes. Remarkably, patients with diabetes mellitus (DM) have a higher incidence and increased severity and risk of complications of bacterial infections [6,7,8]. DM has been identified as a significant predictor of poor prognosis in different bacterial infections and cardiovascular diseases [6,7,8,9,10,11]. The cellular immunity is affected in patients with DM [12,13]. Moreover, decreased activation, chemotaxis, and phagocytosis of polymorphonuclear neutrophils and monocytes are also found in patients with DM [14,15]. Additionally, patients with DM have an increased risk of developing septic shock in different diseases [16,17]. The above reasons may explain why diabetic patients develop unfavorable clinical outcomes in infectious diseases. Some evidences state that strict glycemic control may improve the cellular immunity of patients with DM according to some in vitro and in vivo studies [18,19,20]. More importantly, strict glycemic control can improve the clinical outcomes of patients with critical illness and critical cardiovascular diseases [9,10,11,18,21]. Patients with DM with higher risk of developing IE have been well documented. However, the prognosis of DM with IE is rarely investigated, and results are inconsistent [22,23,24,25]. Therefore, the present study aimed to assess the differences in clinical presentations, microbiology, and in-hospital outcomes between patients with and without DM.

## 2. Materials and Methods

### 2.1. Patient Selection and Study Population

This study was conducted in Kaohsiung Chang Gung Memorial Hospital (Kaohsiung, Taiwan), a 2000-bed tertiary care hospital. We reviewed the medical records of all patients with a discharge diagnosis of IE from February 1999 to June 2012. A total of 412 consecutive patients with IE who were over 18 years old and hospitalized in a tertiary care medical center and who met the modified Duke criteria [26] for the diagnosis of definite IE were included (Figure 1). Data collected for each patient included age, sex, initial presenting symptom, underlying cardiac or other medical illnesses (congenital heart disease, rheumatic heart disease, DM, chronic kidney disease (CKD), liver cirrhosis (LC), malignant diseases, and intravenous drug abusers (IVDA)), history of IE, and preexisting prosthetic valve. The laboratory data included white blood cell count, hemoglobin level, platelet cell count, and renal function test obtained on admission or just before admission. The causative microorganisms were identified by blood culture, and some patients had negative culture result. The echocardiographic findings included the site and size of vegetation, pericardial effusion, valve perforation, and abscess formation. The vegetation length was measured in various planes. In the presence of multiple vegetation sites, the largest length of vegetation was used for analysis. The vegetation length threshold used for analysis was 10 mm [1,27,28]. Neurological, pulmonic, septic, and peripheral embolic events associated with IE were also evaluated.

### 2.2. Definitions

Patients with DM had fasting plasma glucose level of more than 126 mg/dL and glycated hemoglobin A1C level of more than 6.5% during the study period, according to the medical records based on the criteria of guideline [29]. Only patients with transient hyperglycemic episode during hospitalization were diagnosed to have DM. Moreover, LC, IVDA, and history of IE were defined according to the discharge diagnosis found on the medical record. Associated embolic complications were divided into neurological, pulmonary, septic, and peripheral embolic complications. The neurological complications included ischemic stroke, hemorrhagic stroke, mycotic aneurysm with or without cerebral hemorrhage, meningitis, and brain abscess. Patients with transient ischemic attack (duration, less than 24 h) were not included in the group of patients with neurological embolic complications. Peripheral embolic complications included splenic infarction or abscess, renal infarction, Osler’s node, Roth spot, and mycotic aneurysm of the peripheral artery or mesenteric artery. Congestive heart failure (CHF) diagnosed during the initial hospitalization was defined according to the Framingham criteria [30]. The severity of CHF was assessed according to the New York Heart Association (NYHA) Functional Classification. CHF was defined as patients with NYHA functional class III–IV. Age-adjusted Charlson Comorbidity Index (ACCI) was a useful scoring system calculated based on an individual’s age and presence of specified chronic diseases and ACCI was useful scoring tool for predicting outcome in different diseases [31,32].

### 2.3. End Point and Statistical Analysis

The end point of the study was in-hospital mortality. Unless otherwise specified, data are presented as means, standard deviation, or percentages. Continuous variables were compared using the unpaired *t*-test and Wilcoxon rank-sum test, and categorical variables were compared using the chi-squared test. The univariable and multivariable logistic regression analysis were performed to determine the characteristics that were independently associated with in-hospital mortality. The effective sample size is too small the stepwise bootstrap-adjusted analysis including all variables was used to identify best-fitting variables for the final multivariable Cox- regression model [33,34,35]. Clinical, microbiological, and echocardiographic variables found to be significantly associated with mortality in a univariate analysis (*p* < 0.1) were included in the multivariable logistic regression analysis and bootstrap -adjusted analysis. The statistical analyses were performed using a statistical software program (SPSS for Windows, version 17; SPSS Inc., Chicago, IL, USA). Two-sided values of *p* < 0.05 were considered statistically significant. The study (104-6096B) was approved by the Institutional Review Committee on Human Research at Kaohsiung Chang Gung Memorial Hospital.

## 3. Results

### 3.1. Patient Characteristics and Clinical Data

During the 13-year period, a total of 412 patients were diagnosed with definite IE and thus were eligible for this study. The mean age of the patients was 46.11 ± 16.9 years old, and 237 (78.7%) of these patients were male.

Table 1 shows the comparison of the demographic features, underlying diseases, comorbidities, and clinical presentations between patients with DM and without DM. Patients with DM were older compared to those without DM. Additionally, chronic renal failure, hypertension, and prosthetic valve users were more common in patients with DM. High ACCI score was found in patients with DM. IVDA was more frequent in patients without DM than in patients with DM. As regards clinical manifestations during admission, the presence of fever was more frequent in patients without DM compared to patients with DM. The constitutional symptoms including general weakness, low back pain, fatigue, and loss of body weight were more frequent in patients with DM compared to patients without DM. Additionally, the time from admission to the diagnosis of IE was longer in patients with DM compared to patients without DM.

### 3.2. Clinical Laboratory Data, Echocardiographic Findings and Complications of Infective Endocarditis (IE)

The clinical laboratory data, echocardiographic findings, and complications of IE in patients with DM and without DM are shown in Table 2. The hematologic findings were not different between the two groups. However, serum creatinine level was higher in patients with DM compared to patients without DM. Vegetation location on cardiac valves was not different between the two populations. The frequency of cardiac abscess, valve perforation, pericardial effusion, and maximal size of vegetation were not different between the two categories of patients.

The frequencies of neurological complications, pulmonary septic embolisms, peripheral embolic complications, and CHF were not significantly different between the two groups. However, the ratio of IE patients receiving surgical intervention was significantly higher in patients without DM compared to patients with DM.

### 3.3. Causative Microorganism and Complications of IE

Table 3 shows the comparison of microorganism distribution between the two groups of patients. The single significant difference was the higher rate of *Staphylococcus aureus*-related IE in patients with DM (41.7% in patients with DM vs. 27.9% in patients without DM; *p* = 0.021). The frequencies of other microorganism-related IEs were not different between the two groups.

### 3.4. Predictors for In-Hospital Mortality

The in-hospital mortality rate was 21.4% (*n* = 86) and was higher in the diabetic group (41.7%, *n* = 56) compared to the without diabetic group (16.5%, *n* = 30) (*p* < 0.01). In the univariate analysis, the factors associated with in-hospital mortality were age, ACCI score > 3, DM, platelet cell count, *Staphylococcus aureus* infection, neurological embolic complications, and advanced heart failure. (*p* < 0.05, Table 4). Patients with *Viridans streptococcus* infection who received surgical intervention had better prognosis (*p* < 0.05). When these variables were analyzed by multivariable logistic regression and bootstrap-adjusted analysis. DM, age, ACCI score > 3, neurological embolic complications, and advanced heart failure (all *p* < 0.05) were independently associated with in-hospital mortality (Table 5). Additionally, we recalculated the impact of DM in the different periods by performing logistic regression for sensitivity analysis. We found that DM is still an independent predictor of in-hospital mortality (Table 6).

## 4. Discussion

### 4.1. Main Findings

In this study, we analyzed the impact of DM on the in-hospital outcome of IE patients in a 2000 bed-based tertiary hospital. The main findings were as follows: First, this study revealed that patients with DM in our cohort were older and had more comorbid diseases (CKD and HTN) compared to patients without DM. Second, *Staphylococcus aureus* was the most common causative microorganism in patients with DM in the present study. Third, another important finding of this present study was the greater impact of DM as a risk factor of in-hospital mortality in IE patients after adjusting the other confounding factors. The in-hospital mortality rate of this study was 21.4%, which is consistent with the range reported in the previous studies [36,37,38], and it did not decrease despite the availability of advanced diagnostic tools and treatment in the twenty-first century. More importantly, patients with DM had 3.29 times greater in-hospital mortality rate compared to patients without DM after adjusting the other confounding factors.

### 4.2. The Clinical Presentations and Associated Comorbidities of IE Patients with Diabetes Mellitus (DM)

The prevalence rate of DM in hospitalized patients is increasing worldwide [21,25] and the management of complications associated with DM becomes increasingly important. The prevalence rate of DM in this present study is similar to the previous studies, and the prevalence rates of DM with IE are similar in different races. The duration from admission to the diagnosis of IE is longer in the diabetic group in the present study. The clinical characteristics of IE patients with DM were as follows: they were older, had experienced fever less frequently, and were more commonly to experience constitutional symptoms. These characteristics may result in the delay in the diagnosis of IE in patients with DM in our present study. The higher incidence of *Staphylococcus aureus* infection among the microorganisms responsible for IE in patients with DM was found in this present study and is consistent with the previous studies [22,39] A number of probable causes were suggested. First, patients with DM utilize healthcare services more frequently and thus have a higher chance of being exposed to different microoganisms [6,8,17]. Second, diabetic patients have relatively higher risk for skin and mucous membrane bacterial infection compared to nondiabetic patients [6,8,17]. Third, atherosclerotic vascular disease and diabetic neuropathy are predisposing factors in acquiring infections [6,8,17].

### 4.3. The Impact of DM in Predicting In-Hospital Mortality and Probable Mechanisms

The overall in-hospital mortality rate is 21.4% in the present study, which is similar to other countries, including developing countries. However, some controversies surrounding the influence of DM on predicting hospital mortality in IE patients are still present [22,25,39]. Our present study and the other studies [22] found that DM is an independent predictor of in-hospital mortality after adjusting the other confounding factors. There were some reasons for this phenomenon. First, the time from admission to the diagnosis of IE is longer in the DM group compared to the without DM group, which resulted in the delay on the performance of suitable treatment such as administration of antibiotics and/or surgical intervention. The less frequent occurrence of fever in diabetic patients results in the physicians easily ruling out the possible diagnosis of IE. Additionally, the cause of death in patients with DM is sepsis. Real-time and adequate antibiotic treatment is insufficient in these patients; hence, sepsis is the major cause of death in these patients. Second, several studies have found that *Staphylococcus aureus* infection is associated with higher mortality rates in a variety of infectious diseases [40,41,42,43]. The higher rate of *Staphylococcus aureus* infection in diabetic patients conveys an important message, that is, this infection is associated with higher rates of both complications and mortality in IE, and these findings have been confirmed by several studies [42,43,44,45]. Third, the ratio of receiving surgical intervention is lower in the with DM group compared to the without DM group in our study. Previous studies had already found that surgical intervention is a strong protective factor of in-hospital mortality in IE patients [24,46]. Diabetic patients had also been reported to have impaired immune function, hence developing sepsis due to several infectious diseases [47]. Our study also found that patients with DM have a higher rate of mortality due to sepsis compared to patients without DM. Additionally, anatomical complications (e.g., abscess or pericardial effusion) and valve perforation were significantly infrequent in patients with DM. These anatomical complications are absolute indications for surgical intervention in IE patients. [46] These findings may explain the possible causes of poor clinical outcomes in patients with DM, especially complications of sepsis. Recently, some studies demonstrated that early surgical intervention offered benefits to these IE patients [41,48]. A lower surgical intervention rate in patients with DM was found in our study. Early surgical intervention may be considered in IE patients with DM. Finally, ACCI scoring system was used as an independent predictor of mortality in patients with medical conditions and was also found to be an independent predictor of poor outcomes in IE patients. Moreover, our study found that patients with DM were associated with higher ACCI score compared to patients without DM. More importantly, DM is still an independent predictor of in-hospital mortality after adjusting the ACCI score in different periods by multivariable logistic regression analysis.

### 4.4. Clinical Implications

Despite the development of new antibiotic drugs, innovation of bacterial culture techniques, advancement in noninvasive diagnostic imaging modalities, and evolution of modern surgical techniques in the past decades, there is still a lack of improvement in the clinical outcomes of IE patients [3,16,22,25,45]. Therefore, early identification of the risk factors for in-hospital mortality to improve patients’ clinical outcomes is the main goal of the physicians. Our findings found that IE patients with DM had atypical clinical presentation. These phenomena may take longer time in diagnosing IE in patients with DM compared to patients without DM. The physicians should exert their efforts when diagnosing IE especially in patients with DM.

## 5. Study Limitations

This study has several limitations. First, our hospital is a tertiary care center, and some patients coming from community hospitals were transferred in the mentioned hospital. Patient selection biases include greater illness severity and high rates of mortality. Second, all patients underwent transthoracic echocardiography, but transesophageal echocardiography (TEE) was not performed in all patients. However, TEE is not the only diagnostic tool that detects the vegetation in IE patients using the modified Duke criteria [26], and there is no evidence demonstrating that TEE improved the clinical outcome in IE patients. A small number of culture-negative cases may result because of the patients’ pre-exposure to antibiotic treatment, and these patients were transferred from other local hospitals or outpatient clinics; hence, we have no idea as to what types of antibiotics were administered. Third, the duration of our study period is very long, and the quality of patient care may have improved in such a long period of time. Although the sensitivity analysis also found that DM is an independent predictor of in-hospital mortality, we cannot ignore the bias caused by temporal impact. Fourth, this was an observational, small-size study, and all clinical parameters and data were obtained by the retrospective review of medical records. Some parameters like C-reactive protein cannot be included in the analysis due to presence of incomplete data. Finally, the effective sample size is smaller group size in our study. This is barely large enough to include 6-degrees-of-freedom worth of variables in consideration. Although the bootstrap-based variance adjustment [33,34,35] was used to alleviate some bias, this method can’t completely eliminate all bias.

## 6. Conclusions

The mortality rate of IE is still significantly high despite the availability of advanced diagnostic tools and treatment modalities in a tertiary hospital in Taiwan during the twenty-first century. DM is an independent predictor of in-hospital mortality. Therefore, the presence of DM in IE patients should prompt a simple and practical risk assessment.

## Figures and Tables

**Figure 1 ijerph-16-02248-f001:**
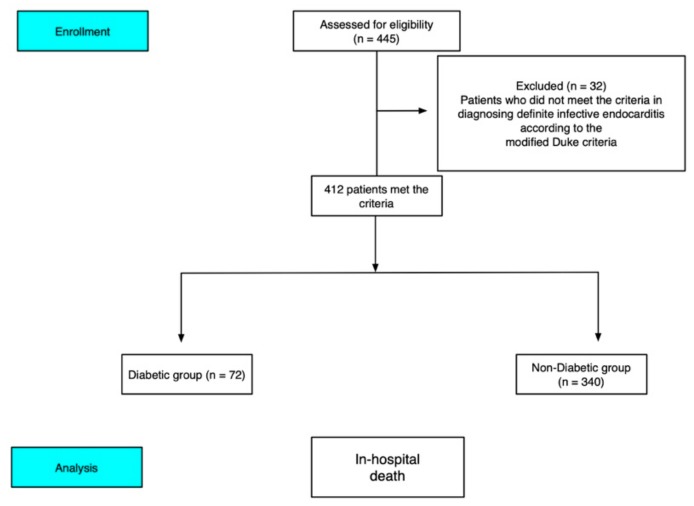
Cohort selection flowchart. The final study cohort included 412 patients diagnosed with definite infective endocarditis according to the modified Duke criteria.

**Table 1 ijerph-16-02248-t001:** Demographic features, underlying diseases, comorbidities, and clinical presentations of 412 patients with infective endocarditis.

Baseline Characteristics	DM (72)	Without DM (340)	*p*
Age	55.9 ± 11.7	44.2 ± 17.1	0.001
Male	75% (54)	79.4% (270)	0.407
Comorbidity			
CHD	6.9% (5)	15.3% (52)	0.06
CKD	26.4% (19)	8.8% (30)	0.001
HTN	51.4% (37)	23.3 (79)	0.001
LC	10.4% (5)	4% (10)	0.099
Malignancy	4.2% (3)	2.4% (8)	0.386
Rheumatic heart disease	11.1% (8)	17.9% (61)	0.159
Prosthetic valve	4.2% (3)	13% (44)	0.033
Previous endocarditis	11.1% (8)	5.9% (20)	0.109
ACCI score	3.24	2.13	0.002
IV drug user	8.3% (6)	28.5% (97)	0.001
Diabetic status			
Oral DM	84.7% (61)	0%	0.001
Insulin DM	15.3% (11)	0%	0.001
Dyspnea	18% (13)	16.2% (55)	0.696
Neurological symptoms	16.7% (12)	13.5% (46)	0.687
Constitutional symptoms	18.1% (13)	7.6% (26)	0.006
Time from admission to the diagnosis of IE (hours)	89 (64–127)	56 (32–89)	0.032

CHD, congenital heart disease; CKD, chronic kidney disease; DM, diabetes mellitus; HTN, hypertension; IV, intravenous; IE, infective endocarditis; LC, liver cirrhosis.

**Table 2 ijerph-16-02248-t002:** Clinical laboratory data, echocardiographic findings, and complications of IE.

	DM (72)	Without DM (340)	*p*
Laboratory findings			
WBC counts (×10^3^/mL)	13.6 ± 6.1	13.9 ± 6.7	0.707
Hemoglobin level (g/dL)	11.1 ± 2.75	10.9 ± 2.33	0.950
Platelet cell count (×10^3^/mL)	185.3 ± 97.8	205.9 ± 112.5	0.235
Serum creatinine level (mg/dL)	2.26 ±2.63	1.52 ± 2.04	0.029
eGFR	41.2 ± 6.5	57.9± 5.4	<0.01
Location of IE			
Aortic valve	44.4% (32)	44.1% (142)	0.676
Mitral valve	40.7% (28)	41.5% (129)	0.880
Tricuspid valve	6.9% (6)	9.4% (32)	0.403
Aortic + mitral valve	2.7% (2)	2.1% (7)	0.705
Echocardiography			
Abscess	2.8% (2)	2.9% (10)	0.940
Valve perforation	5.6% (4)	6.5% (22)	0.772
Pericardial effusion	9.7% (7)	5.3% (18)	0.153
Vegetation *	31.9% (23)	25.3% (86)	0.245
Complication of IE			
Neurological complications	25.1% (18)	22.6% (77)	0.667
Pulmonary septic embolisms	6.9% (5)	13.2 (25)	0.775
Peripheral embolic complications	4.1% (3)	6.7% (23)	0.410
Advanced congestive heart failure	34.7% (25)	27.9% (95)	0.250
Surgical intervention	12.5% (9)	24.4% (83)	0.04
In-hospital mortality	41.7% (30)	16.5% (56)	<0.001

* The maximum length of vegetation > 10 mm; WBC, white cell count; IE, infective endocarditis.

**Table 3 ijerph-16-02248-t003:** Causative microorganisms’ profiles.

Microorganisms	DM (72)	Without DM (340)	*p*
*Staphylococcus aureus*	41.7% (30)	27.9% (95)	0.021
Coagulase-negative *Staphylococci*	8.3% (6)	4.1% (14)	0.131
*Staphylococcus epidermis*	1.4% (2)	4.7% (16)	0.467
*Viridans streptococci*	27.8% (20)	36.2% (123)	0.174
Other *Streptococci* spp.	8.3% (6)	7.4% (25)	0.775
*Enterococcus* spp.	2.8% (2)	6.2% (21)	0.254
Gram-negative bacteria	0% (0)	1.2% (4)	0.355
Fungus	0% (0)	0.3% (1)	0.645
Other microorganisms	1.4% (1)	1.5% (5)	0.958
No microorganism identified	6.9% (5)	10.6% (36)	0.348

**Table 4 ijerph-16-02248-t004:** Univariable logistic regression analysis for the risk factors of in-hospital mortality.

Variables	OR	95% CI	*p* Value
ACCI score > 3	4.42	2.11–6.73	0.002
Ages (per years)	1.09	1.04–1.14	0.011
Male gender	1.08	0.34–1.81	0.781
WBC (increased per 10^3^/mL)	1.00	0.99–1.001	0.418
Serum creatinine level Platelet cell count (increased per 10^3^/mL)	1.06 0.993	0.79–1.32 0.99–0.997	0.5 0.0001
RHD	2.32	0.46–4.18	0.559
Drug abuse	0.96	0.32–1.59	0.417
Previous IE	4.05	0.62–7.47	0.221
Advanced heart failure Liver cirrhosis Diabetes mellitus Chronic renal failure *Viridans streptococci*	8.43 1.26 3.29 1.91 0.36	1.37–15.49 0.57–8.55 1.47–5.11 0.38–3.44 0.16–0.56	0.030 0.261 0.003 0.82 0.02
*Staphylococcus aureus*	3.13	1.15–5.10	0.011
*Enterococci* spp. Neurological complications Surgical intervention	2.73 4.36 0.27	0.79–9.83 2.14–9.17 0.11–0.43	0.12 0.00009 0.004

ACCI, Age-adjusted Charlson Comorbidity index (ACCI); CHD, congenital heart disease; RHD, rheumatic heart disease.

**Table 5 ijerph-16-02248-t005:** Multivariable stepwise logistic regression and bootstrap-adjusted analysis for the risk factors of in-hospital mortality.

Variables	Logistic Regression			Bootstrap-Adjusted		
	OR	95% CI	*p* Value	OR	95% CI	*p* Value
Age	1.08	1.05–1.11	0.001	1.05	1.04–1.06	0.012
ACCI score > 3	3.56	1.89–5.23	0.0032	3.78	1.97–5.59	0.0093
Advanced heart failure	8.76	1.24–16.28	0.041	6.32	2.13–10.51	0.032
Diabetes mellitus	2.36	1.31–3.41	0.012	3.02	1.56–4.48	0.021
*Viridans streptococci*	0.34	0.11–0.57	0.032	0.32	0.13–0.51	0.028
Neurological complications	4.17	2.06–6.28	0.0012	4.52	2.16–6.88	0.0017
Surgical interventions	0.32	0.09–0.55	0.0024	0.29	0.11–0.47	0.0019

**Table 6 ijerph-16-02248-t006:** Calculating the impact of diabetes mellitus for in-hospital mortality in different periods by using univariate and multivariable logistic regression analysis.

Univariable				Multivariable				
Period	Variable	OR	95% CI	*p* Value	Variable	OR	95% CI	*p* Value
1988–2002	DM vs. without DM	2.43	1.31–3.55	0.0012	DM vs. without DM	2.67	1.28–4.06	0.032
2003–2007	DM vs. without DM	4.32	1.91–6.73	0.0047	DM vs. without DM	3.44	1.79–5.09	0.028
2008–2012	DM vs. without DM	3.42	1.56–5.28	0.0058	DM vs. without DM	2.34	1.45–3.23	0.014

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
