# Peer review of "Diabetes Mellitus: An Independent Risk Factor of In-Hospital Mortality in Patients with Infective Endocarditis in a New Era of Clinical Practice"

_ijerph, 2019, doi:10.3390/ijerph16122248_

Round 1

Reviewer 1 Report

In this manuscript titled, " Diabetes Mellitus is still an independent risk factor of in-hospital mortality in patients with infective endocarditis in a new era of clinical practice", Cheng-Jei Lin et al., authors investigate the impact of diabetes mellitus (DM) in long term mortality in patients with infective endocarditis (IE). Authors revealed that the prognosis of IE patients with DM is still poor due to delayed diagnosis, elderly and with more co-morbidities. Early identity and more aggressive treatment may be considered in IE patients with DM.  For the study, the presented data are quite sufficient.

1.     p-value: p should be upper case and italic ‘  P ’, in this manuscript authors use both P and p.

2.     The manuscript should be minor revision with the help of a native English speaker specialist, better the English will be, more chances for international dissemination the paper will have.

Author Response

Dear Reviewer 1:

Your constructive criticism is greatly appreciated.   We have made the following responses to comply with your honorable suggestion.  (Note: The revised parts of the manuscript in response to Reviewer’s comments have been marked in red color):

Question 1: p-value: p should be upper case and italic ‘  P â€™, in this manuscript authors use both P and p.

Response: I apologized for this mistake and fixed this error in new revised manuscript. 

Question 2: The manuscript should be minor revision with the help of a native English speaker specialist, better the English will be, more chances for international dissemination the paper will have

   Response:  The revised manuscript had been edited by a native English speaker and we hope the English language and style of revised manuscript is beyond the standard. Thank reviewer’s suggestion. 

Reviewer 2 Report

The study period is very long -- I would suggest at least doing a sensitivity analysis to check for any temporal impact. You could just add a categorical variable to the model. This should also be acknowledged in the study limitations

Several small grammatical typos throughout the manuscript.

It's not clear if some index of a multiple chronic conditions or overall patient severity was included in the models. Can you clarify. This would be very important in your discussion of diabetes independent contribution to a patient's risk. 

No mention of type of diabetes or severity of diabetes. Do you have a proxy for this -- such as insulin dependence?

Author Response

To Reviewer 2

Thank reviewer’s important criticisms.

Your constructive criticisms are greatly appreciated. We have made the following responses to comply with your honorable suggestion (Note: The revised parts of the manuscript in response to Reviewer’s comments have been marked inred color):

Question 1: The study period is very long -- I would suggest at least doing a sensitivity analysis to check for any temporal impact. You could just add a categorical variable to the model. This should also be acknowledged in the study limitations

Response: Yes, our study period is too long and it’s unknown whether advanced medical therapy alter the outcome of IE.  I performed the sensitivity analysis as reviewer’s suggestion and added table 6 in new revised manuscript.  The results demonstrated diabetes still an independent predictor for in-hospital death in different period. We also add statements in our limitation section of new revised manuscript. 

Question 2: Several small grammatical typos throughout the manuscript.

Response: Yes, there was some small grammatical typos in our original manuscript. We fixed these errors through a native English person editing.  We hope the English language and style of revised manuscript is beyond the standard. Thank reviewer’s suggestion.

Question 3: It's not clear if some index of a multiple chronic conditions or overall patient severity was included in the models. Can you clarify. This would be very important in your discussion of diabetes independent contribution to a patient's risk. 

Response: Age adjusted Charlson Co-morbidity Index(ACCI) was originally used as an independent predictor of death for patients with medical conditions.  Recently, ACCI also has also been proven to be indicator for prognosis of patients with IE.We add ACCI index in our model and found DM is still independent factor for in-hospital death. We thank reviewer’s constructive suggestions again. These data were added in. table 1, 4 and table 5 of our revised manuscript. We also added some discussion about it in our revised manuscript. 

Question 4: No mention of type of diabetes or severity of diabetes. Do you have a proxy for this -- such as insulin dependence?

Response: We divided diabetes as oral or insulin-dependent according to diabetic drugs in table 1 of our new revised manuscript. 

Reviewer 3 Report

In "Diabetes Mellitus is still an independent risk factor of in-hospital mortality in patients with infective endocarditis in a new era of clinical practice," Dr. Cheng-Jei Lin and colleagues conducted an observation study and conclude, "Diabetes Mellitus is still an independent risk factor of in-hospital mortality in patients with infective endocarditis in a new era of clinical practice." Unfortunately, statistical analysis (regression model developing) is flawed in the current form, and the data must be reanalyzed with a statistically sound method. It is highly recommended that the authors consult and include as an author a statistician who understands bias in the method used in this manuscript. All the questions and comments below must be addressed adequately for this manuscript to be considered further. 

Major problems 

The biggest problem is "univariate selection" of the variables used in the multivariable regression model. ("... variables that were found to be significantly associated with death in univariable analysis (p<0.05) were included in the multivariable logistic regression analysis.") This approach is known to induce bias and produce standard error estimates that are too small and exaggerate the statistical significance. The variables in the multivariable model must be identified prior to investigating their association with the response. Because this association has already been studied, there may by no way to avoid bias. Perhaps, there may be some way to estimate and correct for such a bias using a bootstrap-related methodology. 

Number of variables in the model is another concern. The effective sample size is the number of the smaller group (events/non-events), and it is 86 in this study. This sample size allows 5 ~ 7 variables (degrees of freedom) to be examined. Note that this is not the number of variables in the final model, but the number of variables examined. In other words, this data set is too small for the conducted statistical analysis.

The collected data represent 13 years of clinical practice, and the quality of patient care must have changed (improved) in such a long period of time. This needs to be accounted for in the analysis. 

With the model that was used in this version, the main conclusion, "The prognosis of IE patients with DM is still poor due to delayed diagnosis, elderly and with more co-morbidities." does not follow. There are no data or analysis that support "due to ..." part of the conclusion. 

MInor points

In table 4, for the continuous variable, indicate what changes in the variable (e.g., platelet cell count) that the odds ratio is correspond to. 

Please show actual p-values, not just "p<0.05" or "p<0.01".

"Allocation" in Figure 1 is misleading given that this is an observational study.

There are a number of misspelled words. (e.g., then -> than, platelet)

"Multivariate analysis" is a wrong term. It should be "Multivariable analysis"

Author Response

Responses Reviewer's Specific Comments (Reviewer 3):

Dear Reviewer 3:

Your constructive criticism is greatly appreciated.   We have made the following responses to comply with your honorable suggestion (Note: The revised parts of the manuscript in response to Reviewer’s comments have been marked inblue color):

Major problems 

Question 1: The biggest problem is "univariate selection" of the variables used in the multivariable regression model. ("... variables that were found to be significantly associated with death in univariable analysis (p<0.05) were included in the multivariable logistic regression analysis.") This approach is known to induce bias and produce standard error estimates that are too small and exaggerate the statistical significance. The variables in the multivariable model must be identified prior to investigating their association with the response. Because this association has already been studied, there may be no way to avoid bias. Perhaps, there may be some way to estimate and correct for such a bias using a bootstrap-related methodology. 

Number of variables in the model is another concern. The effective sample size is the number of the smaller group (events/non-events), and it is 86 in this study. This sample size allows 5 ~ 7 variables (degrees of freedom) to be examined. Note that this is not the number of variables in the final model, but the number of variables examined. In other words, this data set is too small for the conducted statistical analysis.

Response: Thank reviewer’s constructive suggestion. Yes, there were some problems in our initial manuscript about the selection of univariable factor into multivariable regression model. The clinical, microbiological, and echocardiographic variables found to be significantly associated with mortality in a univariate analysis (< 0.1) were included in the multivariate logistic regression analysis. Therefore, we re-calculate the univariate/multivariable regression model and add the results by using bootstrap-related methodology in our new revised manuscript. These data make our manuscript more solid and readable. We rewrote the statistic method section and re-calculated the table 4 and table 5 in our revised manuscript.  The second problem is about the number of variables in our manuscript. Our event number (in-hospital death) is 86 and sample size is 412.  In fact, the number in some manuscripts about the predictor of mortality in patients with IE is also 300-400 case. We think the number in our study is enough for analysis but the number is still a limitation of our study.  We add this limitation in the limitation section.

Question 2: The collected data represent 13 years of clinical practice, and the quality of patient care must have changed (improved) in such a long period of time. This needs to be accounted for in the analysis. 

Response:  The duration of our study is a long period and the quality of patient care may change(improved) in such a long period of time.   I performed the sensitivity analysis and added table 5 in new revised manuscript.  The results demonstrated diabetes still an independent predictor for in-hospital death in different periods.  The results reflect diabetes is still an independent predictor for in-hospital mortality in different period. 

Question3: With the model that was used in this version, the main conclusion, "The prognosis of IE patients with DM is still poor due to delayed diagnosis, elderly and with more co-morbidities." does not follow. There are no data or analysis that support "due to ..." part of the conclusion.

Response: We apologized the inadequate description about "The prognosis of IE patients with DM is still poor due to delayed diagnosis, elderly and with more co-morbidities. We re-wrote the statement ‘The prognosis of IE patients with DM is still poor’ in our new revised manuscript. 

Question 4: In table 4, for the continuous variable, indicate what changes in the variable (e.g., platelet cell count) that the odds ratio is correspond to. 

Answer: We re-wrote the presentation style in table and believe it is more suitable for reader. 

Question 5:Please show actual p-values, not just "p<0.05" or "p<0.01".

Answer: We re-wrote the actual p value. 

Question 6"Allocation" in Figure 1 is misleading given that this is an observational study.

Answer: We apologized t misleading given allocation. Thank reviewer’s suggestion again.

There are a number of misspelled words. (e.g., then -> than, platelet)

Response: Yes, there was some misspelled words. We had fixed the errors in our new manuscript by a native English-editing person. 

Question 7 : "Multivariate analysis" is a wrong term. It should be "Multivariable analysis"

Response: We had fixed the errors in our new manuscript.

Finally, we again thank the reviewers for their comments. 

Round 2

Reviewer 3 Report

 "Our event number (in-hospital death) is 86 and sample size is 412.  In fact, the number in some manuscripts about the predictor of mortality in patients with IE is also 300-400 case."

The effective sample size in logistic regression is the smaller group size (86). This is barely large enough to include 6-degrees-of-freedom worth of variables in consideration (not in the final model, but in consideration). The boot strap-based variance adjustment alleviates some but not all. This needs to be more clearly stated in the limitation section. 

Please describe the bootstrap adjusted analysis in more detail with appropriate citations because it is not yet a common approach that every reader is familiar with. 

Author Response

Responses Reviewer's Specific Comments (Reviewer 3):

Dear Reviewer 3:

Your suggestion is greatly appreciated.  We have made the following responses to comply with your honorable suggestion (Note: The revised parts of the manuscript in response to Reviewer’s comments have been marked in red color):

Question 1

Response: Thank reviewer’s constructive suggestion. 

Yes, the effective sample size is smaller in our study. Although we used the bootstrap adjusted analysis to correct the bias, the boot strap-based variance adjustment alleviates some but not all. We added these statements in our limitation section and described the bootstrap adjusted analysis in method section with appropriate citations. We hope these changes makes manuscript for suitable reading for reader.  

Thank you again.
